# A multidimensional functional fitness score has a stronger association with type 2 diabetes than obesity parameters in cross sectional data

**Pramod Patil[1], Poortata Lalwani[2], Harshada Vidwans[3], Shubhankar Kulkarni[1], Deepika Bais[3], Manawa Diwekar-Joshi[3], Mayur Rasal[1], Nikhila Bhasme[1], Mrinmayee Naik[1], Shweta Batwal[1], Milind Watve** [1] *

**1** BILD Clinic, Deenanath Mangeshkar Hospital, Pune, India, **2** Department of Psychology, University of Michigan, Ann Arbor, MI, United States of America, **3** Department of Biology, Indian Institute of Science Education and Research, Pune (IISER–P), Pune, India

* milind.watve@gmail.com

**Data Availability Statement:** All relevant data are within the paper and its Supporting Information files.

## Abstract

### Objectives

We examine here the association of multidimensional functional fitness with type 2 diabetes mellitus (T2DM) as compared to anthropometric indices of obesity such as body mass index (BMI) and waist to hip ratio (WHR) in a sample of Indian population.

### Research design and method

We analysed retrospective data of 663 volunteer participants (285 males and 378 females between age 28 and 84), from an exercise clinic in which every participant was required to undergo a health related physical fitness (HRPF) assessment consisting of 15 different tasks examining 8 different aspects of functional fitness.

### Results

The odds of being diabetic in the highest quartile of BMI were not significantly higher than that in the lowest quartile in either of the sexes. The odds of being a diabetic in the highest WHR quartile were significantly greater than the lowest quartile in females (OR = 4.54 (1.95, 10.61) as well as in males (OR = 3.81 (1.75, 8.3). In both sexes the odds of being a diabetic were significantly greater in the lowest quartile of HRPF score than the highest (males OR = 10.52 (4.21, 26.13); females OR = 10.50 (3.53, 31.35)). After removing confounding, the predictive power of HRPF was significantly greater than that of WHR. HRPF was negatively correlated with WHR, however for individuals that had contradicting HRPF and WHR based predictions, HRPF was the stronger predictor of T2DM.

### Conclusion

The association of multidimensional functional fitness score with type 2 diabetes was significantly stronger than obesity parameters in a cross sectional self-selected sample from an Indian city.

**Funding:** The authors received no specific funding for this work.

**Competing interests:** The authors have declared that no competing interest exists.

**Abbreviations:** BMI, Body Mass Index; WHR, Waist hip ratio; HRPF, Health related physical fitness; T2DM, Type2 diabetes mellitus; OR, Odds Ratio; NS, Non Significant.

## Introduction

Contrary to the classical belief, it is being increasingly clear that actual evidence for obesity being the main, central and causal factor for insulin resistance and type 2 diabetes mellitus (T2DM) is quite weak and debated [1]. Many shortcomings and paradoxes in this view are becoming apparent with increasing research. (i) The direction of causality between obesity, insulin resistance and hyperinsulinemia is debated. While the classical view presumes obesity to be primary, giving rise to insulin resistance and hyperinsulinemia as a compensatory response of the body, an upcoming view is that hyperinsulinemia has a causal role in obesity [2–5]. Downregulation of insulin expression by insulin gene dosage [6], pharmaceutical suppression of insulin [7] or fat cell specific insulin receptor knockouts [8] protect against obesity. Epidemiological evidence is predominantly associative and does not clearly show causal relationship of obesity with insulin resistance and hyperinsulinemia. (ii) The association between obesity and insulin resistance is quantitatively quite weak throughout the globe with the median variance explained being only about15% across published studies [9]. Classes of individuals that are normal weight but metabolically obese [10] and obese but metabolically normal [11] are well known and are not rare by any standard. Individual metabolic responses to a given level of obesity have a large variance the reasons for which are not clearly known [12]. (iii) Further, the association varies between populations, normal weight T2DM being more common in south Asians [13, 14]. (iv) As a result of the emphasis on obesity, little guidelines for preventive interventions for the normal weight metabolically obese individuals are available [10].

The definition of obesity and the reliability of measurable indices is also debated. Body mass index (BMI) is perhaps the most commonly used index, but its limitations have been well recognized [15]. Estimates of total body fat do not consistently improve the association with insulin resistance than the anthropometric measures across studies [9, 16, 17]. The distribution of fat is claimed to be different in the metabolically healthy obese [12]. However, why the distribution of fat varies between individuals is not clearly known. It is certainly not explained by energy balance alone and therefore there is little guidance for effective intervention to improve fat distribution.

Given the limitations of obesity and its anthropometric indicators as predictors of T2DM, one needs to look for alternative and more reliable predictors. There have been arguments and evidence about fitness being more important than fatness [18–25]. However, most studies have looked at a single dimension of fitness such as grip strength [21, 22], lower body strength [24, 25] or cardiorespiratory fitness [18–21]. Most of these studies find that some form of functional fitness has a protective role against T2DM and many other life-style related disorders. However, fitness is a multidimensional concept which is difficult to capture in a single task performance test. Whether a multidimensional fitness score has greater predictive power than a single dimension or single task has not been addressed seriously. Currently there is no standardized set of test protocols for assessing multidimensional functional fitness. We found an opportunity to examine multidimensional fitness score as a predictor of T2DM using retrospective data from an exercise clinic in Pune, India, which routinely conducted a multidimensional fitness test at entrance for every new participant. We used this opportunity to make a preliminary assessment of whether a comprehensive index of fitness gives us a cross-sectional predictability substantially greater than that given by obesity parameters. Despite the limitations of being a retrospective study, the results are suggestive of a major conceptual change.

## Methods

### The clinic and the patient group

Deenanath Mangeshkar Hospital and Research Centre, a multi-speciality hospital in Pune city, India, opened an exercise clinic namely BILD (behavioural intervention for lifestyle disorders)

clinic in 2017. It serves as an exercise training centre for prevention as well as treatment for neuro-orthopedic, metabolic and endocrine problems. The exercise prescription is personalized depending upon the problem addressed along with the capacity and physical limitations of the individual. In order to record the capacity and limitations of the patients, the centre conducts a routine health related physical fitness (HRPF) examination of every entrant. HRPF is a multidimensional functional fitness test consisting of 15 small tasks that examine different dimensions of physical fitness including abdominal plasticity, balance, endurance, flexibility, nerve-muscle coordination, muscle strength, core strength and agility. The set, described in the [S1 File], gives a differential score to each of the fitness components and a total composite fitness score. The HRPF protocol has been standardized at this clinic to serve the purpose of assessing the physical capacity of an entrant before prescribing any exercise regime. During past one year the clinic had performed over 800 HRPF assessments covering age groups between 18 and 84. We noted that type 2 diabetics in the sample ranged between ages 28 and 84 and therefore we used the same range for the non-diabetic counterpart of the sample. This resulted in the selection of 285 males and 378 females out of which 81 males and 69 females were type 2 diabetic (Table 1).

The nature of the study was retrospective. The study proposal was reviewed by the Institutional Ethics Committee, In-house Research, of the DeenanathMangeshkar Hospital and Research Centre. The committee approved the study and owing to the retrospective nature, the requirement for informed consent was waived by the committee and the data were anonymized before availing for analysis which was done between October to December 2018.

## Statistics used

We used Pearson's correlation to examine the interrelationship between different fitness components and their relationship with age and obesity parameters. Bonferroni correction was applied to use a conservative significance level since multiple correlations were being performed. Since T2DM is an age related disorder and since age had a significant negative correlation with fitness, age corrected fitness scores were used for further analysis.

The male and female populations were divided into quartiles according to HRPF, individual fitness components, BMI and WHR. We looked at the odds ratios in males and females separately as well as in the pooled data. For pooling male female data, males and females were

**Table 1. The age sex characteristics and the mean HRPF score with and without age correction.**

|  |  | Diabetic | Non-diabetic |
|---|---|---|---|
| Males | Sample size | 81 | 204 |
|  | Age range | 33–84 | 28–80 |
|  | Age mean (stddev) | 59.91 (10.74) | 47.38 (11.99) |
|  | BMI (stddev) | 26.2 (3.45) | 26.89 (3.64) |
|  | WHR (stddev) | 0.976 (0.049) | 0.947 (0.046) |
|  | HRPF score mean (stddev) | 46.71 (15.37) | 63.91 (12.8) |
|  | Age corrected HRPF | 49.92 (8.95) | 103.4 (10.68) |
| Females | Sample size | 69 | 309 |
|  | Age range | 28–76 | 28–80 |
|  | Age mean (std dev) | 56.43 (10.6) | 44.49 (11.06) |
|  | BMI (std dev) | 28.37 (4.23) | 28.2 (5.15) |
|  | WHR (std dev) | 0.922 (0.067) | 0.87 (0.07) |
|  | HRPF score mean (std dev) | 39.95 (13.56) | 55.5 (13.04) |
|  | Age corrected HRPF (std dev) | 42.66 (8.02) | 89.14 (11.01) |

ranked separately, quartiles identified separately and then pooled together. Odds of being a diabetic were calculated for each of the quartiles. Odds ratio (OR) between the 1st and 4th quartiles was used as a cross sectional predictor. For the functional fitness components the odds of being diabetic were expected to be higher in the lowest quartile therefore the ratio of first to forth quartile was expressed. In the case of BMI and WHR, odds in the fourth quartile were expected to be higher and therefore the ratio was expressed as forth to first quartile. The significance of difference between odds ratios was tested by z transformation of log OR difference using the standard errors of OR.

To examine whether the best or the worst fitness components contributed more to the association between HRPF and T2DM, we ranked individuals for each functional fitness component independently. We then added the 3 highest and 3 lowest ranking scores of every individual and looked at odds of being diabetic across their quartiles.

## Results

The age distribution of the diabetics in both sexes was negatively skewed and that in the non-diabetics was positively skewed. Therefore although we matched the age range, the mean age of diabetics was greater than that for non-diabetics in both the sexes (Table 1). Therefore for all further analysis we take every variable after correcting for age according to the regression in the data pooled for diabetics and non-diabetics.

BMI did not correlate with age significantly in either of the sexes (males r = -0.136, NS; females r = 0.085, NS). In both sexes WHR correlated positively with age (males r = 0.287, p < .001; females r = 0.227, p<0.001). In males, the HRPF score decreased with age but the rate of decrease was significantly steeper in diabetics ($r^2$ = 0.451 and regression slope = – 0.961, CI = – 0.72 to– 1.28) than non-diabetics ($r^2$ = 0.342, slope = – 0.625, CI = – 0.505 to– 0.74). The pattern was similar in females but the difference in the slopes was not significant (diabetics $r^2$ = 0.342 and slope = – 0.747, CI = – 0.531 to –1.03)) (non-diabetics $r^2$ = 0.296, slope = – 0.642, CI = – 0.523 to– 0.7451)). Nevertheless the best fit regression line for diabetics lay below that for non-diabetics throughout the age range indicating that at any age group diabetics were functionally less fit than non-diabetics (Fig 1).

The different components of functional fitness were intercorrelated positively but the $r^2$ values were small, over 95% of them ranging between 0.00001 and 0.25 with occasional outliers going up to 0.42 (Fig 2). The components were significantly positively correlated to the total score as expected in both sexes and the component that explained maximum variance in HRPF was muscle strength (63.2% in males and 40.3% in females).

After correcting for age, HRPF score was not correlated with BMI either in diabetics or non-diabetics in both sexes. But BMI was negatively correlated with some of the components of functional fitness namely abdominal plasticity (r = – 0.187, p <0.001) and core strength (r = – 0.247, p < 0.001) in females, balance (in males r = – 0.323, p < 0.001 and in females– 0.430, p < 0.0001) and muscle strength (in males r = – 0.233, p< 0.001; in females r = – 0.409 p <0.001) in both sexes. WHR was significantly negatively correlated with age corrected HRPF (males r = – 0.231, p< 0.001; females r = – 0.239, p< 0.001). Although statistical significance was seen, the coefficients of determination in both the cases were very small (0.052 and 0.057). BMI was not significantly correlated with WHR in both sexes. Thus neither BMI nor WHR reflected on functional fitness very well.

The odds of finding a diabetic in the lowermost quartile of BMI was significantly greater than that in the highest quartile, contrary to the expectation in males (OR = 0.41 (0.19, 0.89); but not in females OR = 0.65 (0.31, 1.37)) (Fig 3). The odds of being a diabetic in the highest WHR quartile were significantly greater than the lowest quartile in females (OR = 4.54 (1.95,

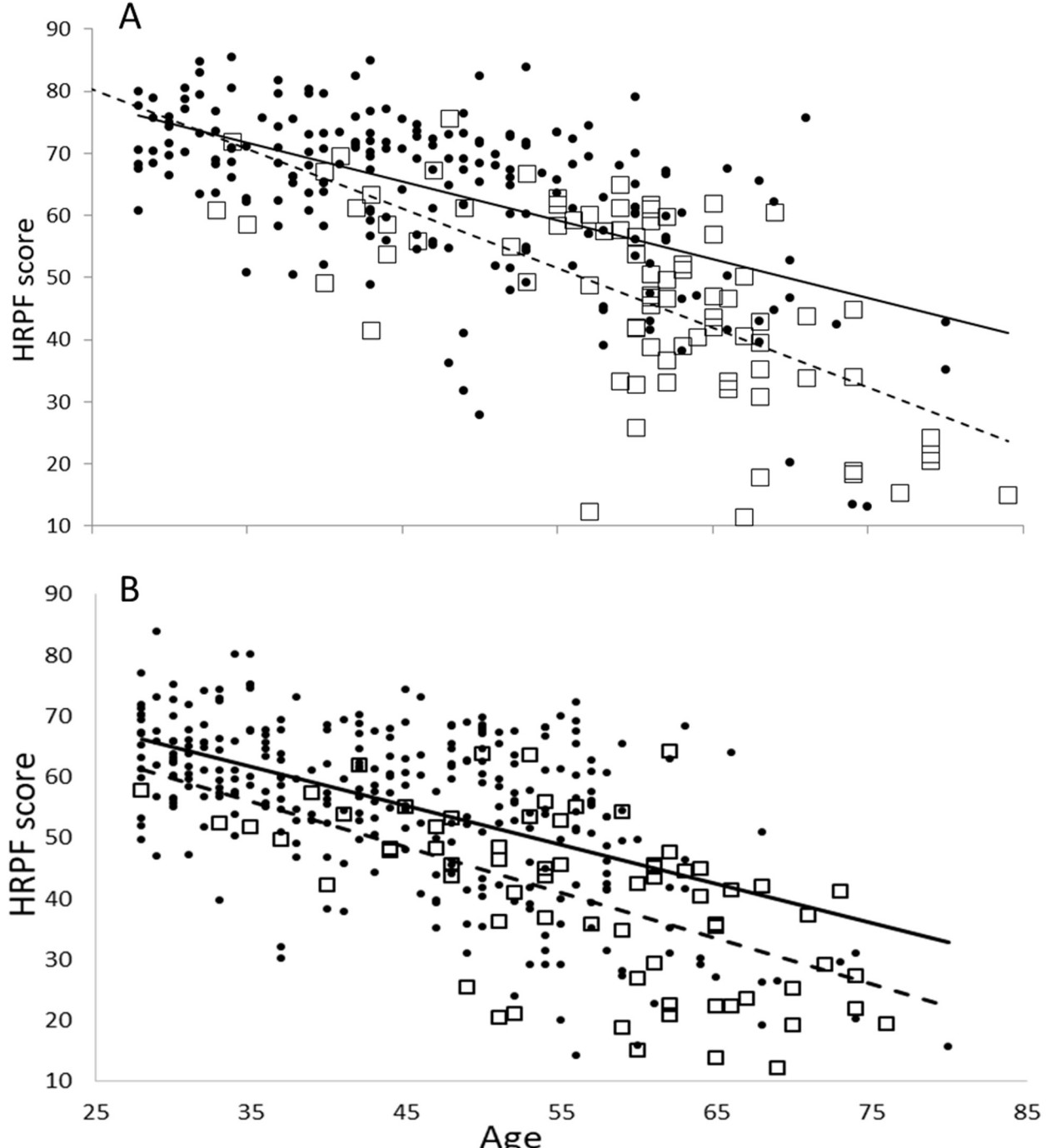

**Fig 1.** Age trend in HRPF scores in diabetics (dark circles) and non-diabetics (hollow squares) with respect to HRPF scores (A) males (B) females.

10.61)) as well as in males (OR = 3.81 (1.75, 8.3)). In both sexes the odds of being a diabetic were significantly greater in the lowest quartile of HRPF score than the highest (males OR = 10.52 (4.21, 26.13); females OR = 10.50 (3.53, 31.35)). In both sexes the ORs for HRPF were over two fold those of the corresponding ORs for WHR but the differences were not statistically significant. However when data on the two sexes were pooled with nested ranking,

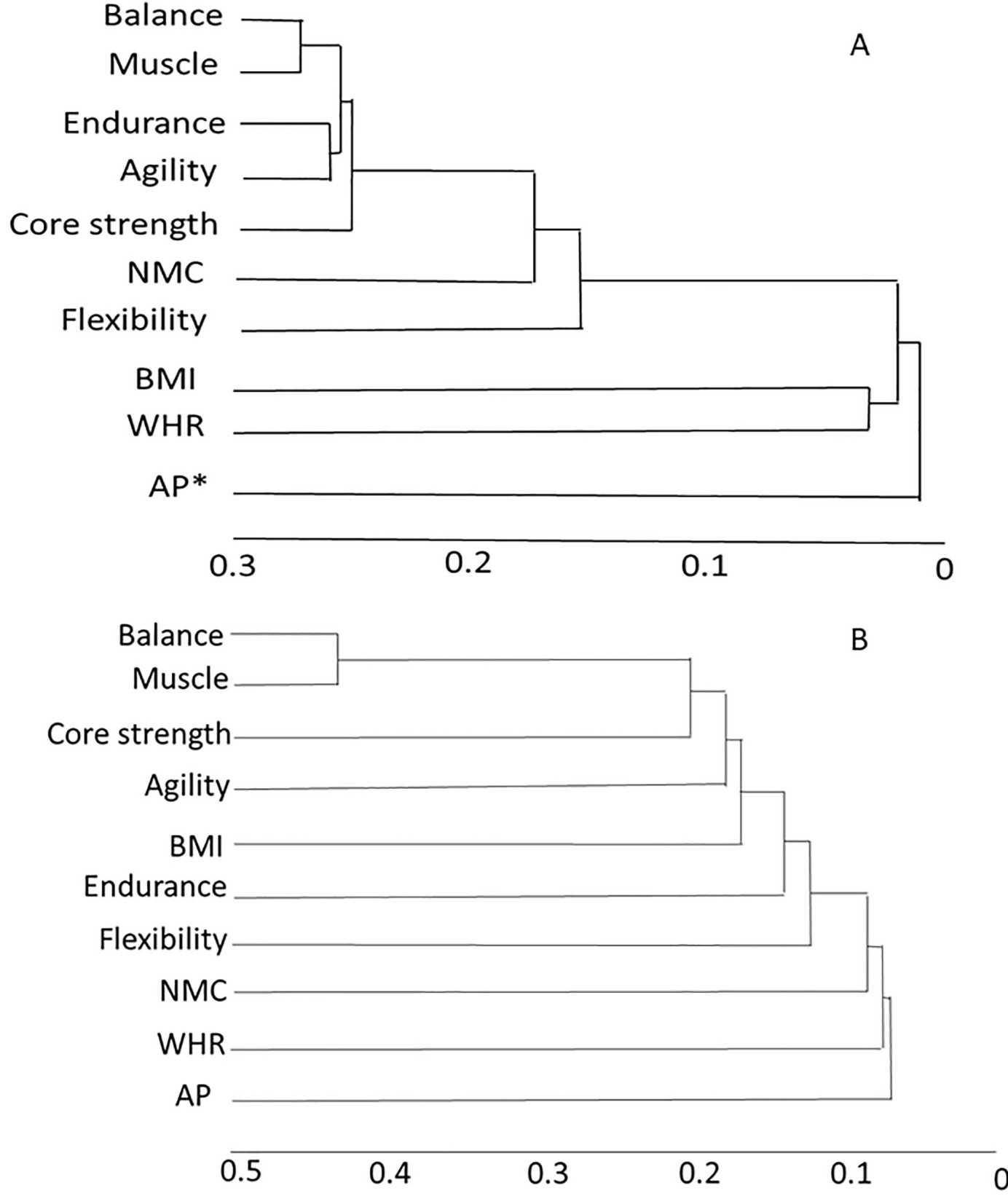

**Fig 2.** (A) males (B) females: clustering of different components of functional fitness based on the pair wise coefficients of determination, using nearest neighbor clustering. Note that most coefficients are below 0.3 with the exception of balancing and muscle strength in females. Thus no single component represents overall fitness very well.

the OR for HRPF (OR = 9.78 (4.93, 19.8)) was significantly greater than the OR for WHR (4.03 (2.09, 7.08); p one tailed = 0.03). Therefore the predictive power of HRPF appears to be greater than WHR and BMI.

The odds ratios of the lowest to highest quartiles of some of the individual fitness component scores (Fig 3) were also significant. However the ORs of individual fitness components were lower than that for the total HRPF score and not significantly different from each other (Fig 4). The only exception was nerve-muscle coordination in males (OR = 16.63 (6.36, 35.6)) for which the OR was greater than the OR for HRPF but the difference was not significant. However it was significantly greater than the OR for WHR (p one tailed 0.019). The predictive power of the multidimensional functional fitness generally appeared to be better than most of the single task scores or single dimensions of functional fitness.

Since WHR and HRPF were good predictors of diabetes but not BMI, we tested whether WHR and HRPF were independent predictors. After correcting for WHR, HRPF remained a significant predictor although the odds ratio decreased in males (OR = 5.05 (2.33, 10.98)) but increased in females (OR = 22.3 (6.59, 75.5)). Reciprocally after correcting for HRPF, WHR remained a significant predictor with a marginal decrease in odds ratios (males 3.766 (1.7, 8.37); females (2.58 (1.18, 5.63)). This indicates that both WHR and HRPF are independent predictors. Moreover after removing the confounding the OR for HRPF in females was significantly greater than OR for WHR (p one tailed 0.0015). Also with both sexes pooled, the difference was statistically significant (p one tailed = 0.0068). Thus with or without confounding, multidimensional functional fitness score appears to have a greater predictive value than the obesity parameters BMI and WHR.

Since WHR and HRPF are independent predictors of T2DM, one would expect maximum risk for individuals with unfavorable fitness score as well as unfavorable WHR. Simultaneously it would be enlightening to see the risk for individuals that have favorable functional fitness but unfavorable WHR, or favorable WHR but unfavorable functional fitness. Since the correlation between WHR and HRPF was weak, it was possible to find individuals in the fourth quartile of WHR which is most unfavorable for health but the most favorable (forth) quartile of HRPF and vice versa. When the proportion of diabetics in the four combinations were plotted (Fig 5) the maximum incidence (32 out of 59) was in the most unfavorable HRPF as well as most unfavorable WHR classes as expected, and lowest (only 2 out of 44) in the diametrically opposite combination. This indicates that the effects of WHR and HRPF are synergistic. However it can be seen that the gradient across HRPF was much sharper than the gradient across WHR. Furthermore in the combination where WHR was unfavorable and HRPF was favorable, the incidence was only 1 out of 18, but when HRPF was unfavourable and WHR was favourable it was 12 out of 31. So when cases in which the two predictors were in opposite direction, HRPF was a better quantitative predictor than WHR.

In order to examine whether the association between lower HRPF scores and T2DM is dominated by the weaker components of fitness or the stronger ones, we ranked all individuals according to each of the component fitness scores and then for each individual totaled the lowermost and uppermost three ranks. The index thus obtained for the weaker dimensions of fitness was a much better predictor of T2DM, the odds ratio for males being OR = 13.4 (4.84, 37.14) and for females OR = 19.51 (4.47, 84.3). On the other hand the index for the three strongest components had a relatively lower predictive power, OR for males being 8.52 (3.06,

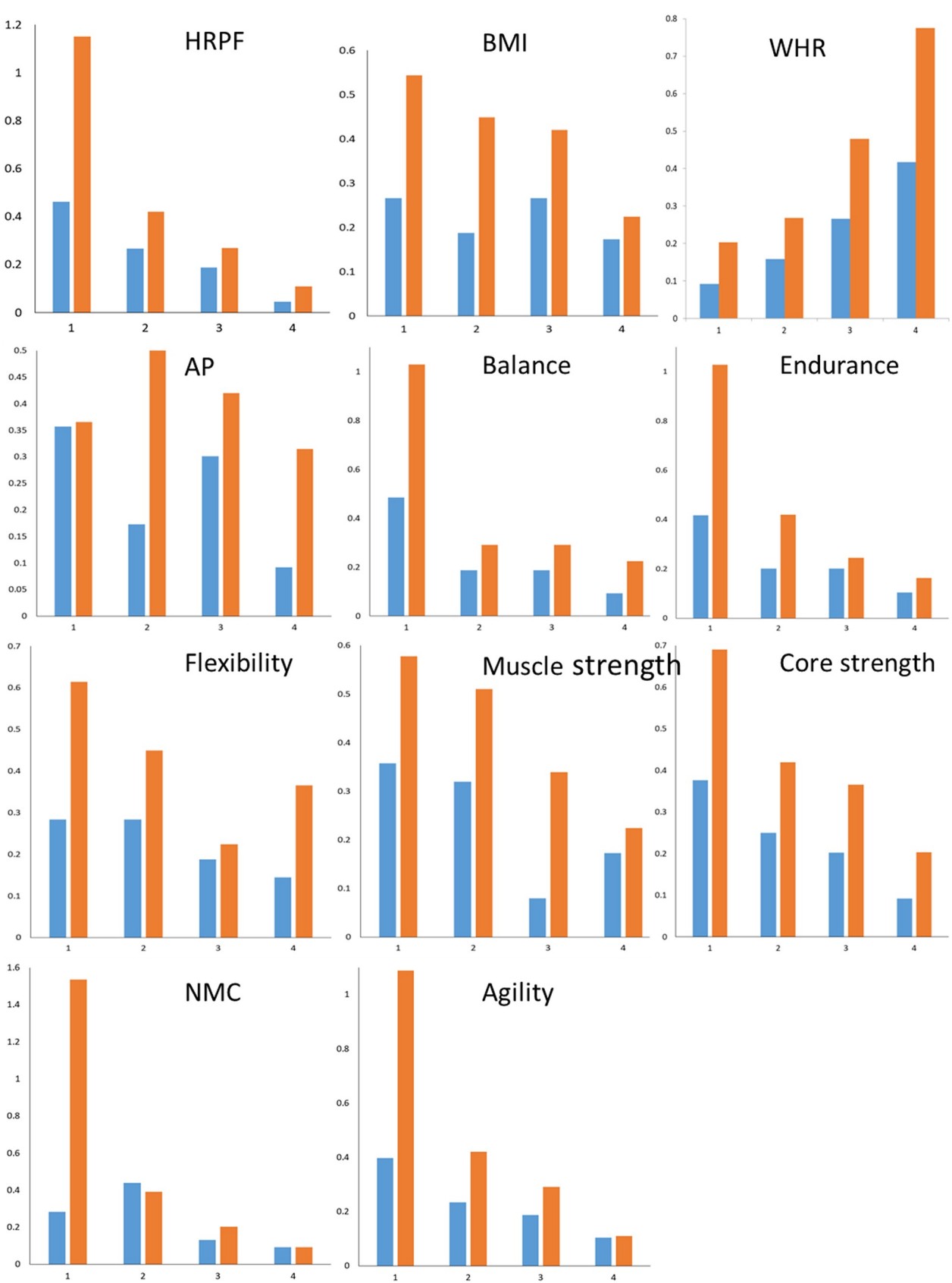

**Fig 3. Odds of being type 2 diabetic among quartiles of different components of functional fitness in comparison with morphometric indices.** Males- orange bars, females–blue bars.

23.76); females 4.64 (2.06, 10.4)). The difference between ORs of weakest and strongest components was statistically significant for females (p one tailed 0.039), for males it was not statistically significant but the direction of change was similar. This suggests that weakest fitness dimensions appear to influence the association more than the strongest fitness dimensions for any individual.

Since we observed from the data that there was a significant association between low fitness score and T2DM, we tried to address the question whether diabetes leads to progressive

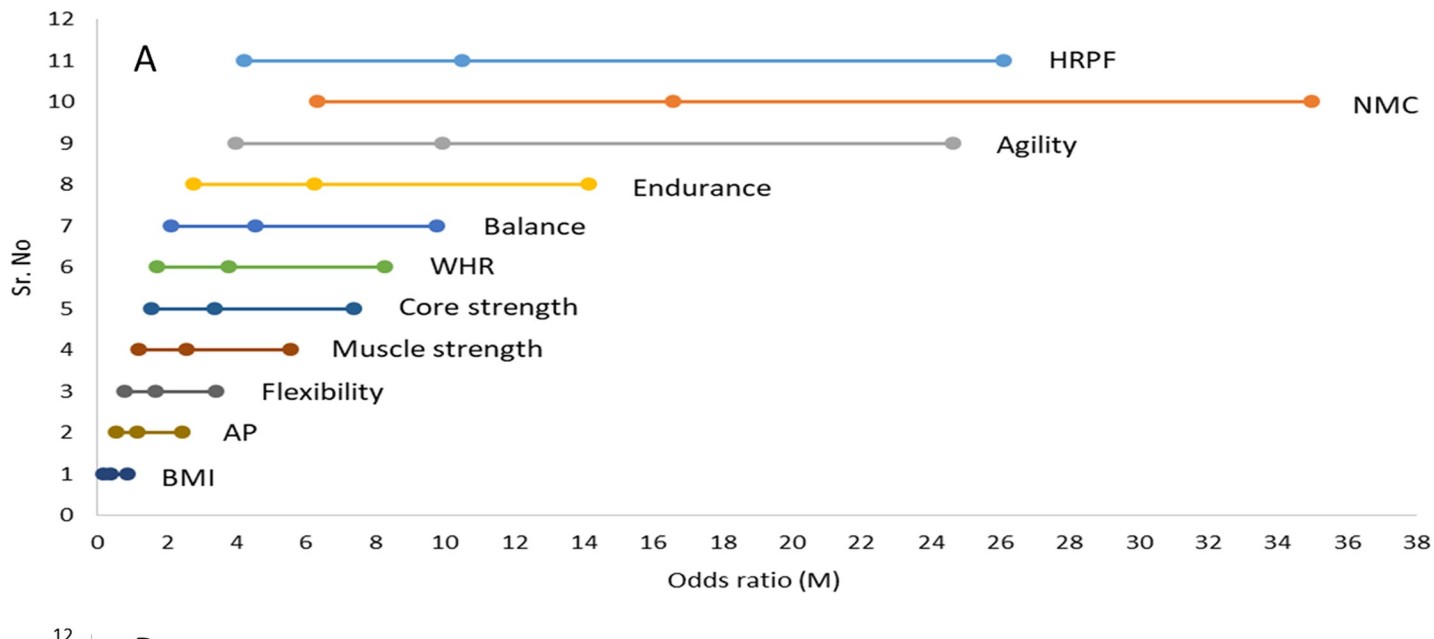

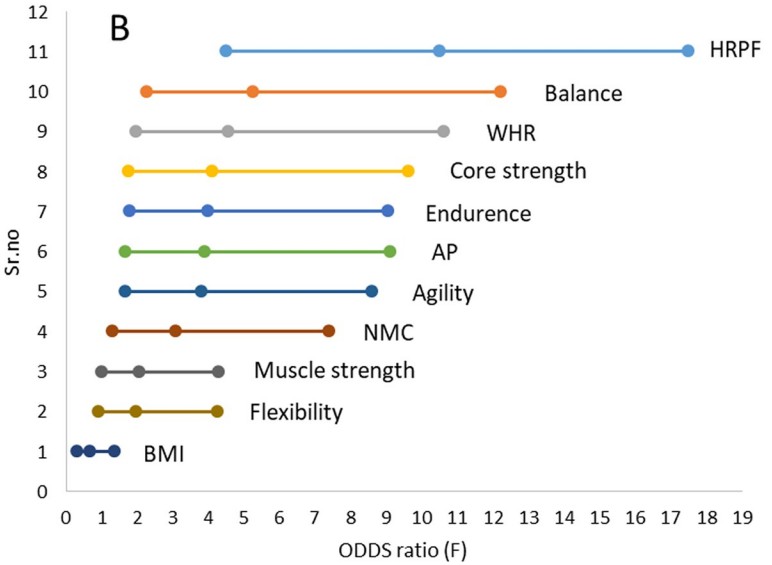

**Fig 4. The ORs across first and 4th quartile for being diabetic and their confidence intervals for different fitness dimensions, collective HRPF and morphometric parameters.** A: males, B: females.

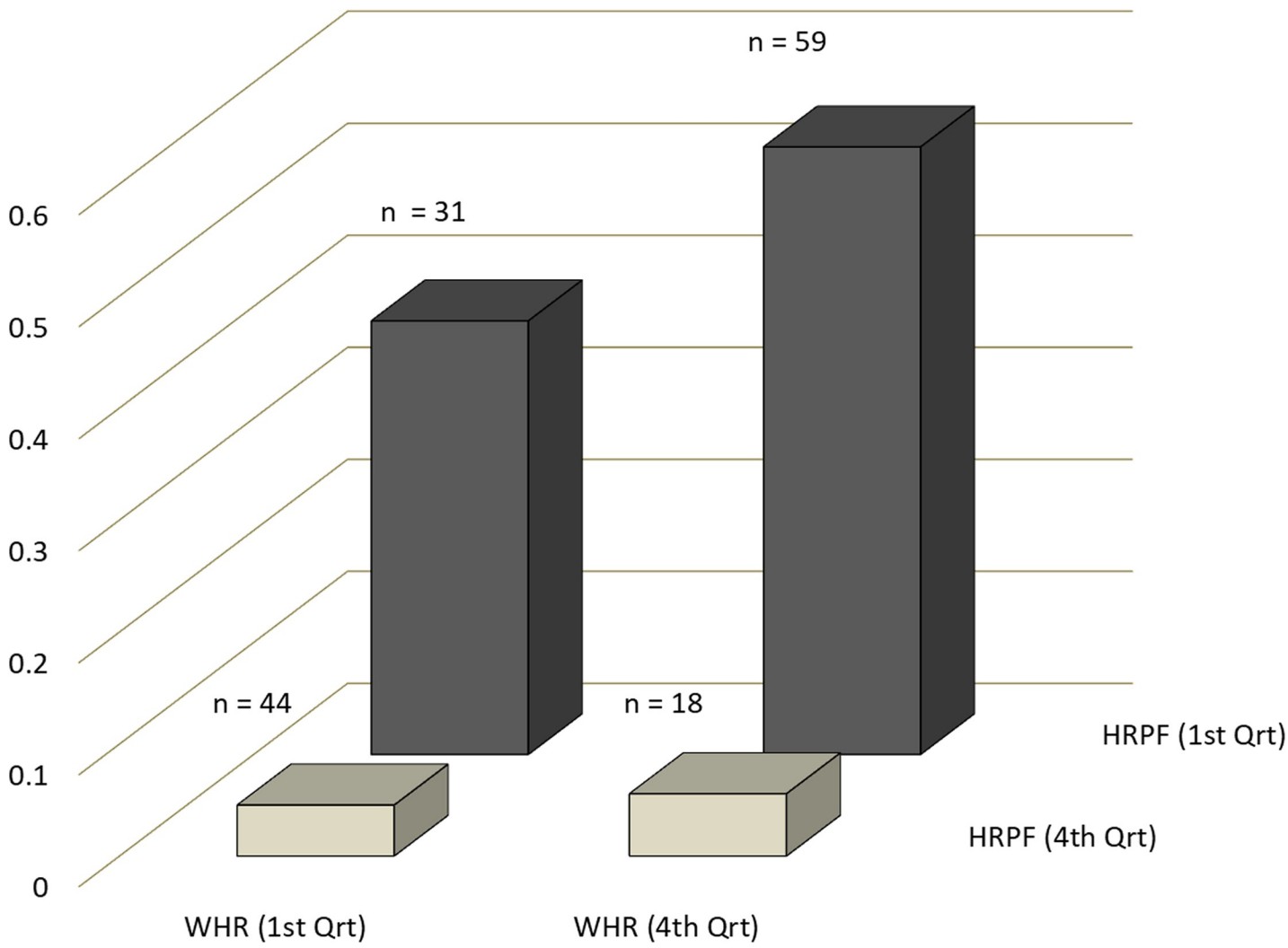

n = 59

n = 31

n = 44

n = 18

HRPF (1st Qrt)

HRPF (4th Qrt)

0.6
0.5
0.4
0.3
0.2
0.1
0

WHR (1st Qrt)

WHR (4th Qrt)

**Fig 5. The proportion of diabetic people along the first and forth quartiles of WHR and HRPF.** As expected, the proportion in the lowest HRPF and highest WHR quartiles is the largest and highest HRPF and lowest WHR quartiles the smallest. Particularly notable pattern is that individuals that have good HRPF scores had low incidence even when WHR was bad. On the other hand when WHR was good but HRPF bad, the incidence was substantially higher. Pooled data on both sexes where ranking is done separately in males and females.

decline of fitness or whether loss of fitness was a predisposing factor for diabetes. The age related decline in HRPF was faster in diabetics than in non-diabetics in males but not in females as shown earlier. If diabetes leads to progressive loss of fitness, we would expect people with longer duration of diabetes to have lower fitness scores. Among the individuals in the sample, data on duration of diabetes was available for 52 males and 38 females. In this subsample, we did find a significant negative correlation between HRPF and duration of diabetes in males ($r = -0.38$, $p = 0.01$). However, after correcting for age the correlation was lost. On the contrary, the age-HRPF negative correlation was not lost after correcting for duration of diabetes (corrected $r = -0.569$, $p = 0.01$). This suggests that the apparent negative correlation of HRPF with duration of diabetes is likely to be contributed by age alone. In females, on the other hand, there wasn't a significant correlation between HRPF and duration of diabetes ($r = -0.237$, $p > 0.05$). Therefore the prediction of the hypothesis that diabetes leads to a progressive loss of fitness was not supported by data. Although the causal relation cannot be

ascertained from this analysis alone, its inability to support progressive effect of diabetes on fitness makes it more likely that loss of fitness predisposes to diabetes.

## Discussion

The central finding of the study is that the multidimensional fitness score was a substantially better predictor of T2DM as compared to BMI and WHR in a cross sectional sample of the Indian population. Our study was retrospective and was not a randomized sample from the population. Nevertheless there is no apparent reason why a self-selection bias would lead to lower fitness scores in diabetics as compared to non-diabetics. The study is certainly indicative and should be followed up with prospective studies with better sampling design. The HRPF protocol used was not intended to be a predictor of diabetes. But since we find it to be of substantial potential interest, it might be possible to refine, standardize and validate the index further for use as a diabetes risk factor. Thus although our study has the limitations of a retrospective study, the possible implications can be potentially important for the prevention of type 2 diabetes.

The different dimensions of functional fitness such as balance, endurance, flexibility, nerve-muscle coordination, muscle strength, core strength and agility captured by component fitness scores were correlated positively with each other but the correlations were weak. This means that the different components of fitness are unlikely to be represented well by a single test or measurement. Fitness is necessarily a multidimensional concept and only anthropometry or a single task performance does not reflect on overall fitness sufficiently well. In particular, BMI and WHR were poorly correlated with the functional fitness components and therefore are unlikely to be good surrogates of overall fitness. Among individuals with an impairment such as osteoarthritis, anthropometric indices may have a stronger correlation with loss of functional fitness [26, 27] but these associations in terms or OR or HR are substantially weaker than those between HRPF and T2DM found in this study [28, 29]. Therefore it is unlikely that function fitness association with T2DM is because of obesity.

The question whether any particular single component of fitness is a better predictor of T2DM needs to be kept open since we did not find a significant difference in their predictive ability in terms of OR. Nevertheless almost all single components had ORs smaller than the total score which indicates the importance of multidimensional functional fitness score. If some component is found to be consistently the best predictor across different populations, it may become a simple and single useful test in future, but from our sample it appears that it is necessary to look collectively at different dimensions of fitness.

The exercise of taking the lowermost and uppermost scores suggests, on the other hand, that rather than a single component, an individual's weakest fitness components appear to determine the risk of being diabetic. This means that a balanced and all round fitness needs to be emphasized and only being normal weight or strong in one or two components may not be sufficient. Different types of exercises have differential emphases on particular fitness components. Therefore rather than one particular type of monotonous exercise, diversity of exercises strengthening different components of functional fitness is likely to be a more successful preventive measure. This is an interesting possibility raised by this study which needs more research to explore its translational importance.

The study exposes the extremely limited role of obesity parameters in T2DM either as causal or simply correlated variables. Particularly since south Asia is known to have a substantial number of normal weight type 2 diabetics, it is necessary to move the focus from obesity to better correlated, and if possible demonstrably causal factors. Sarcopenia is a known predisposing factor for insulin resistance [30, 31] which would be reflected in some components of

the fitness tests. However in our study balance, endurance and nerve muscle coordination were also good predictors of T2DM without necessarily being highly correlated with muscle strength. Therefore it is possible that apart from fat and muscle mass, a number of other functional fitness parameters play a role. Other signals such as EGF, BDNF, FGF, other growth factor signals [32–37], autonomic neuronal signals [38], myokines [39–41], muscle damage, pain and infection [42–44] have been shown to affect insulin sensitivity partly or completely independent of obesity. Therefore it is necessary to look beyond obesity parameters as risk factors and potentially causal factors.

Why should low functional fitness be associated with T2DM? In evolutionary medicine a number of novel hypotheses for the origin of T2DM have been suggested. The classical thrifty gene [45] and thrifty phenotype [46] hypotheses are obesity centered. There are alternative hypotheses based on behavioural and reproductive strategies or life-history strategies [47–49] which are not obesity centered, although they might allow a correlation with obesity. In these hypotheses, physical strength, social ranking and thereby reproductive opportunities play a more important role than obesity. A definite role of brain and neuronal circuits is being highlighted by a number of recent studies [50–52]. In an ancestral environment, physical fitness is expected to play a role in deciding behavioural and reproductive strategies and therefore a change in metabolic and neuroendocrine make up may follow loss of physical fitness. For example, a physically weak individual is most likely to be a subordinate individual in a primate social hierarchy and accordingly needs to change its foraging, social and reproductive strategies [53, 54]. The position in social hierarchy might be lost by the loss of one or more components of fitness. Here the weaker components are likely to matter more than the stronger components. The strategic changes required on being weak are accompanied by metabolic, endocrine and immunological fine tuning [52–55]. Although the relationship between physical fitness and social hierarchy has changed in the modern human society, human physiology is still likely to be responding according to the evolved ancestral optimization. This is the potential theoretical underpinning of the significant association between functional fitness and T2DM.

Our finding, if confirmed across populations, has important clinical implications. While the prevention and treatment for obesity is based on improving energy balance, the treatment does not specifically address fat distribution. On the other hand specific exercises can be designed to improve the deficient components of fitness specifically. Such a strategy needs to be explored with further focused research.

## Conclusion

In a retrospective study of the Indian population loss of functional fitness was found to have a significantly greater association with type 2 diabetes than the anthropometric parameters of obesity or central obesity. Functional fitness is a multidimensional concept and the association of the multidimensional fitness score appeared to be stronger than most of the individual fitness components. The study indicates that loss of any of the functional dimensions might contribute to this association. Although this study does not ascertain the causal relationship, we did not find support to the hypothesis that diabetes progressively led to loss of fitness, keeping the possibility open that loss of fitness may predispose to diabetes.

## Declarations

Ethics approval and consent to participate: This was a retrospective study with data available on the clinic record, so consent of participants is not relevant.

Consent for publication: All authors and the Hospital authorities have their consent to publish.

Availability of data and materials: Details of the HRPF assessment and the datasets analyzed during the current study are available from the corresponding author on reasonable request.

## Supporting information

**S1 File. Protocol for conducting and scoring HRPF.**
(DOCX)

**S1 Data.**
(XLSX)

## Author Contributions

**Conceptualization:** Pramod Patil, Poortata Lalwani, Deepika Bais, Milind Watve.

**Data curation:** Harshada Vidwans.

**Formal analysis:** Harshada Vidwans, Shubhankar Kulkarni, Milind Watve.

**Methodology:** Pramod Patil, Poortata Lalwani, Shubhankar Kulkarni, Deepika Bais, Manawa Diwekar-Joshi.

**Project administration:** Pramod Patil, Mayur Rasal, Nikhila Bhasme, Mrinmayee Naik, Shweta Batwal.

**Visualization:** Poortata Lalwani.

**Writing – original draft:** Pramod Patil, Harshada Vidwans, Milind Watve.

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
