## [Decision Letter · Decision Letter 0]

27 Oct 2020

PONE-D-20-22351

A multidimensional functional fitness score has a stronger association with type 2 diabetes than obesity parameters in cross sectional data

PLOS ONE

Dear Dr. Watve,

Thank you for submitting your manuscript to PLOS ONE. After careful consideration, we feel that it has merit but does not fully meet PLOS ONE’s publication criteria as it currently stands. Therefore, we invite you to submit a revised version of the manuscript that addresses the points raised during the review process.

We look forward to receiving your revised manuscript.

Kind regards,

Antonio Palazón-Bru, PhD

Academic Editor

PLOS ONE

Journal Requirements:

'The study proposal was reviewed by the Institutional Ethics Committee, In-house

Research, of the Deenanath Mangeshkar Hospital and Research Centre on 27th Aug

2018. Owing to the retrospective nature, the requirement for informed consent was

waived by the committee and the data were anonymized before availing for analysis

which was done between October to December 2018.'                                       

a. Please amend your current ethics statement to confirm that your named institutional review board or ethics committee specifically approved this study.     

b. Once you have amended this/these statement(s) in the Methods section of the manuscript, please add the same text to the “Ethics Statement” field of the submission form (via “Edit Submission”).For additional information about PLOS ONE ethical requirements for human subjects research, please refer to http://journals.plos.org/plosone/s/submission-guidelines#loc-human-subjects-research'

"We thank the management of Deenanath Man geshkar Hospital and Research Center that supports the BILD clinic, Mr. Deepak Desai and Mrs. Sarita Desai for financing the study..."

"The authors received no specific funding for this work."

Reviewers' comments:

Reviewer's Responses to Questions

**Comments to the Author**

1. Is the manuscript technically sound, and do the data support the conclusions?

Reviewer #1: Yes

2. Has the statistical analysis been performed appropriately and rigorously? 

Reviewer #1: Yes

3. Have the authors made all data underlying the findings in their manuscript fully available?

Reviewer #1: Yes

4. Is the manuscript presented in an intelligible fashion and written in standard English?

Reviewer #1: Yes

5. Review Comments to the Author

Reviewer #1: Abstract

- No comments

Introduction

- The first sentence is unclear. What is the main point?

- The argument is not clearly set up. It seems that the authors argue in the first paragraph that a causal relationship between obesity and T2DM does not exist. The shift in the second paragraph to measuring fitness is abrupt.

- The authors contend that anthropometric measures of obesity are not as relevant as fitness, but no papers discussing how obesity relates to anthropometric measures are discussed in the introduction. It would be helpful for the authors to cite the research measuring anthropometrics in adults with obesity to explain why they feel that fitness is a more relevant measure (see below).

o The association of waist circumference with walking difficulty among adults with or at risk of knee osteoarthritis: the Osteoarthritis Initiative

SV Gill et al., Osteoarthritis and cartilage 25 (1), 60-66

- Overall, the introduction needs to be reworked to clearly state what the argument is.

Method

- What do the authors mean by abdominal plasticity?

- Although the authors state what areas of fitness the HRPF measures, they do not provide any details about what the 15 tasks actually are.

-

Results

- No comments

Discussion

- A significant limitation of this study is that there is no measure of percent body fat. WHR provides a rough estimate of the distribution of body fat, but a formal measure of body fat, skeletal muscle mass, etc. would provide a clearer picture about why the authors argue the limited role of anthropometric measures in understanding limitations related to obesity.

6. PLOS authors have the option to publish the peer review history of their article (what does this mean?). If published, this will include your full peer review and any attached files.

Reviewer #1: No

---

## [Author Response · Author response to Decision Letter 0]

28 Nov 2020

Dear Editor,

Thanks for giving us the opportunity to revise. 

Please find below our specific responses (in blue) to individual issues. 

'The study proposal was reviewed by the Institutional Ethics Committee, In-house

Research, of the Deenanath Mangeshkar Hospital and Research Centre on 27th Aug

2018. Owing to the retrospective nature, the requirement for informed consent was

waived by the committee and the data were anonymized before availing for analysis

which was done between October to December 2018.' 

a. Please amend your current ethics statement to confirm that your named institutional review board or ethics committee specifically approved this study. 

b. Once you have amended this/these statement(s) in the Methods section of the manuscript, please add the same text to the “Ethics Statement” field of the submission form (via “Edit Submission”).For additional information about PLOS ONE ethical requirements for human subjects research, please refer to http://journals.plos.org/plosone/s/submission-guidelines#loc-human-subjects-research'

The ethics statement is modified accordingly and corresponding change in the form made. 

"We thank the management of Deenanath Man geshkar Hospital and Research Center that supports the BILD clinic, Mr. Deepak Desai and Mrs. Sarita Desai for financing the study..."

"The authors received no specific funding for this work."

The funding mentioned earlier was not for this study, but for raising the clinic infrastructure. That mention in the acknowledgement is removed now. So the funding statement remains unchanged. 

 Data availability statement remains unchanged and will make the repository address available on acceptance. 

 Rechecked and found appropriate.

5. Review Comments to the Author

Reviewer #1: Abstract

- No comments

Introduction

- The first sentence is unclear. What is the main point?

- The argument is not clearly set up. It seems that the authors argue in the first paragraph that a causal relationship between obesity and T2DM does not exist. The shift in the second paragraph to measuring fitness is abrupt.

The opening paragraph is reorganized in the revised MS for greater clarity. 

- The authors contend that anthropometric measures of obesity are not as relevant as fitness, but no papers discussing how obesity relates to anthropometric measures are discussed in the introduction. It would be helpful for the authors to cite the research measuring anthropometrics in adults with obesity to explain why they feel that fitness is a more relevant measure (see below).

We have added a new paragraph in the introduction section with references to discuss the association and possible causation links between fat distribution, anthropometry and metabolic effects. 

o The association of waist circumference with walking difficulty among adults with or at risk of knee osteoarthritis: the Osteoarthritis Initiative

SV Gill et al., Osteoarthritis and cartilage 25 (1), 60-66

Thanks for bringing this paper to our notice. Citing a few other papers along with this we have added a para in the discussion section that discusses the association between anthropometric parameters and functional fitness. 

- Overall, the introduction needs to be reworked to clearly state what the argument is.

We hope in the revised and rewritten form the introduction has enhanced clarity and makes the purpose of the study clear. 

Method

- What do the authors mean by abdominal plasticity?

- Although the authors state what areas of fitness the HRPF measures, they do not provide any details about what the 15 tasks actually are.

All details of the 15 tasks including abdominal plasticity are described in details in the supplementary information, which is cited in the main text in the methods section. The reviewer may have missed it. 

-

Results

- No comments

Discussion

- A significant limitation of this study is that there is no measure of percent body fat. WHR provides a rough estimate of the distribution of body fat, but a formal measure of body fat, skeletal muscle mass, etc. would provide a clearer picture about why the authors argue the limited role of anthropometric measures in understanding limitations related to obesity.

In the paper describing the retrospective part of the study, data on body composition was not available. This can be included in a prospective study. Nevertheless we have added in the revised paper along with selected references a brief discussion on whether body fat is a better predictor of insulin resistance or T2DM.

---

## [Decision Letter · Decision Letter 1]

22 Dec 2020

A multidimensional functional fitness score has a stronger association with type 2 diabetes than obesity parameters in cross sectional data

PONE-D-20-22351R1

Dear Dr. Watve,

We’re pleased to inform you that your manuscript has been judged scientifically suitable for publication and will be formally accepted for publication once it meets all outstanding technical requirements.

Kind regards,

Antonio Palazón-Bru, PhD

Academic Editor

PLOS ONE

Additional Editor Comments (optional):

Reviewers' comments:

Reviewer's Responses to Questions

**Comments to the Author**

1. If the authors have adequately addressed your comments raised in a previous round of review and you feel that this manuscript is now acceptable for publication, you may indicate that here to bypass the “Comments to the Author” section, enter your conflict of interest statement in the “Confidential to Editor” section, and submit your "Accept" recommendation.

Reviewer #1: All comments have been addressed

2. Is the manuscript technically sound, and do the data support the conclusions?

Reviewer #1: Yes

3. Has the statistical analysis been performed appropriately and rigorously? 

Reviewer #1: Yes

4. Have the authors made all data underlying the findings in their manuscript fully available?

Reviewer #1: Yes

5. Is the manuscript presented in an intelligible fashion and written in standard English?

Reviewer #1: Yes

6. Review Comments to the Author

Reviewer #1: The authors have satisfactorily responded to the reviewers' comments. I think that the manuscript is ready to be accepted.

7. PLOS authors have the option to publish the peer review history of their article (what does this mean?). If published, this will include your full peer review and any attached files.

Reviewer #1: No

---

## [Editor Report · Acceptance letter]

25 Jan 2021

PONE-D-20-22351R1 

A multidimensional functional fitness score has a stronger association with type 2 diabetes than obesity parameters in cross sectional data 

Dear Dr. Watve:

I'm pleased to inform you that your manuscript has been deemed suitable for publication in PLOS ONE. Congratulations! Your manuscript is now with our production department. 

Kind regards, 

on behalf of

Dr. Antonio Palazón-Bru 

Academic Editor

PLOS ONE